# Are Adherence to the Mediterranean Diet, Emotional Eating, Alcohol Intake, and Anxiety Related in University Students in Spain?

**DOI:** 10.3390/nu12082224

**Published:** 2020-07-25

**Authors:** Marchena Carlos, Bernabéu Elena, Iglesias M. Teresa

**Affiliations:** 1Faculty of Education and Psychology, Universidad Francisco de Vitoria, 28223 Madrid, Spain; 2Faculty of Health Sciences, Universidad Francisco de Vitoria, 28223 Madrid, Spain

**Keywords:** Mediterranean diet, KIDMED, emotional eating, alcohol abuse, audit, anxiety, college students

## Abstract

Research has suggested that university students are at risk from certain unhealthy habits, such as poor diet or alcohol abuse. At the same time, anxiety levels appear to be higher among university students, which may lead to high levels of emotional eating. The aim of this study was to analyze the degree of adherence to the Mediterranean diet (AMD), emotional eating, alcohol intake, and anxiety among Spanish university students, and the interrelationship of these variables. A total of 252 university students filled out the Mediterranean Diet Quality Index (KIDMED) questionnaire for Mediterranean diet adherence, an Alcohol Use Disorders Identification Test, a State-Trait Anxiety Inventory and the Emotional Eater Questionnaire. We analyzed descriptive data, a *t*-test and analysis of variance (ANOVA) for differences, a Pearson correlation, and multiple regression tests. Results showed low levels of AMD among university students (15.5%) and considerable levels of emotional eating (29%) and anxiety (23.6%). However, levels of alcohol dependence were low (2.4%). State-anxiety was a predictor of the emotional eater score and its subscales, and sex also was predictive of subscale guilt and the total score. However, AMD was predicted only by trait-anxiety. These models accounted for between 1.9% and 19%. The results suggest the need for the implementation of educational programs to promote healthy habits among university students at risk.

## 1. Introduction

College students find themselves in a transition period towards adulthood in which they have reached biological maturity without comparable psycho-social development [1]. The beginning of the university experience is usually accompanied by significant changes in lifestyle, including changes in diet and poor eating habits such as skipping meals, inadequate nutrition, and frequent intake of fast food [2,3]. This unhealthy behavior is associated, among Mediterranean populations, with a departure from the Mediterranean dietary model, defined as being low in saturated fat and sweets, and high in olive oil, vegetables, fruits, cereals, nuts, and legumes [4], while poor nutrition is generally associated with substance abuse including the excessive intake of alcohol [5,6]. A review of studies carried out between 2002 and 2014 on the habits of university students found that, on the whole, students do not have healthy eating habits, have a poor diet with a high intake of calories and excessive alcohol consumption, among other substances. These findings confirm the results of recent studies which show a wide range of high-risk behaviors among university students, including excessive alcohol consumption and poor eating habits [7,8,9]. In contrast, adherence to the Mediterranean Diet (AMD) is associated with better cognitive performance and emotional wellbeing [10]. The degree of adherence of Spanish university students to MD has been the subject of a number of studies [2,11,12] which have used the Mediterranean Diet Quality Index (KIDMED) questionnaire [13]. Findings show that a lower degree of adherence is related to problems of mental health and higher levels of negative emotions, such as stress, anxiety, and, consequently, lower self-esteem [14]. By contrast, greater adherence to MD combined with greater socialization reduces the risk of depression by some 50% [10,15].

Sex seems to be a determining factor in food patterns and alcohol intake; better nutritional profiles in woman have been found than in men in a sample of 987 student from the University of Balearic Islands (Spain) [16], and it has been widely reported that women drink less and have a lower prevalence of drink problems than men [17], although recent studies suggest a significant increase in problematic alcohol consumption among women [18]. Furthermore, although older students drink less and less excessively than younger students [19], there is wide cross-country variation; age was positively associated with alcohol consumption among Chinese college students, but negatively related to alcohol consumption frequency among Germans [17].

Emotional states can influence the quantity and quality of food consumption [20]. Studies conducted in recent decades have found that stress and negative emotions are often associated with increased food consumption [21], unless the person is in a highly intense emotional state (tension, panic) in which case appetite can disappear entirely [22]. Anxiety can also lead to greater consumption of foods with a high fat and sugar content as a coping strategy [22] and may alter the cerebral reward system, changing healthy dietary habits in favor of unhealthy foods [23].

The concept of emotional eating refers to the tendency to eat as an automatic response to negative emotions [24] and implies the disproportionate intake of food, regardless of the hunger the person may feel [25]. This type of intake is related to feelings of anxiety and depression among the general population [26], and with certain sex differences: among men emotional eating has been principally associated with symptoms of depression while in women it is primarily associated with anxiety [27]. Contrarily, other authors have argued that, among the non-clinical population, eating is often a response to boredom rather than anxiety or depression [25]. Whatever benefits the strategy of emotional eating may afford in dealing with anxiety, these tend to be short-lived, often followed by other negative emotions such as feelings of guilt [28]. Among the clinical population, this type of behavior shows a high rate of comorbidity (74%), with Axis 1 DSM IV (Diagnostic and Statistical Manual of Mental Disorders fourth edition) disorders (anxiety disorders and mood disorders) [29].

University students also show higher levels of anxiety than the general population [30] and worry about academic performance appears to be the main driver of this distress [31,32]. Stress, anxiety and depression appear to be the principal psychological problems experienced during university studies all over the world and studies show university students have poorer mental health than the general population [33,34]. A study of Spanish university students observed that some 44.7% of students showed emotional distress indicative of anxiety, regardless of their field of study [35]. The absence of effective strategies to deal with situations of stress may result in the adoption of inappropriate or harmful coping mechanisms such as alcohol consumption or unhealthy eating habits. A recent study evaluating the frequency of emotional eating among 335 female university students during a period of 28 days found that half reported experiencing these episodes at some moment, and 51.3% of these were associated with anxiety [36]. An inverse relationship was also found between alcohol consumption and the frequency of these episodes, in line with data which suggest that the intake of alcohol may replace emotional eating as a coping strategy for negative emotions, especially anxiety [37].

Regarding alcohol consumption, young adults generally drink to excess as a way to reduce stress and deal with social anxiety [38]. Studies of alcohol consumption among university students show that over 50% of students report consuming alcohol on weekends and that the pattern of consumption known as “binge drinking”, the episodic intake of large quantities of alcohol in a short period of time, is an increasing public health issue with predicable consequences [39,40]. The patterns of consumption appear to differ according to sex; as opposed to men, emotional distress and depression among women has an inverse correlation to alcohol consumption [41].

Given this scenario, it would appear important to assess unhealthy habits and explore more deeply the relationship between eating habits, alcohol consumption, and anxiety among students, particularly among future health professionals such as physiotherapy, psychology, and nursing students, who will be responsible for promoting health in the general population in the near future. Understanding the role of anxiety in poor diet and excessive alcohol intake will provide information that will enable one to develop and implement the appropriate prevention and intervention strategies to ensure the adoption of healthy habits among the health care professionals in training and among college students in general. On the other hand, it seems important to assess the level of anxiety of the students of different health degrees in order to develop coping strategies in situations of distress that do not involve unhealthy habits.

The objectives of the present study are:To analyze the levels of anxiety, emotional eating, alcohol consumption, and AMD of university students enrolled in health sciences degrees and explore the differences between students of physiotherapy, psychology, and nursing.To study the relationship between alcohol consumption, AMD, emotional eating, and anxiety among Spanish university students in health sciences.To determine if age, sex, and anxiety are predictors of alcohol consumption, emotional eating, and adhesion to the Mediterranean diet.

## 2. Materials and Methods

### 2.1. Participants

The sample consisted of 252 university students from the Francisco de Vitoria University in Madrid, Spain. The students were enrolled in different degree programs in the area of health sciences: nursing, physiotherapy, and psychology. For convenience, a non-aleatory sample was used. Participation was voluntary and confidential, and all participants signed an informed consent form. Given the interest of our study in the sub-clinical population, we used any psychological diagnosis and drugs consumption as exclusion criteria. Table 1 provides an analysis of the age of the participants by sex and area of university study.

### 2.2. Variables and Instruments

Socio-demographic variables: an ad hoc questionnaire was used to collect information on different socio-demographic variables such as age, sex, place of origin, degree studies, etc.

Adherence to the Mediterranean diet: the KIDMED test was used, a brief instrument for dietary evaluation providing information on the AMD [13]. The test consists of 16 dichotomous questions in which participants must specify if they perform the indicated task. The items aim to assess the intake of food and habits belonging to the Mediterranean diet (fruit, olive oil, cheese, etc.) and the consumption of inadequate food contrary to Mediterranean diet (sweets, ultra-processed, fast food, etc.) The test provides a global score used to obtain an index classifying the participants by poor diet, need to improve dietary habits, and optimum Mediterranean diet, as specified in thedata analysis section. The reliability of this test was verified in a validation test with Spanish children and young people up to 24 years of age [42].

Alcohol consumption: The Alcohol Use Disorder Identification Test (AUDIT) was used to determine the alcohol consumption habits of the participants. The questionnaire was created by the World Health Organization to identify persons with high-risk or harmful alcohol consumption habits by means of a simple screening test. The test consists of 10 items, 8 of which are on a Likert scale of 5 categories ordered from 0 (never/1 or 2 units) to 4 (daily/10 or more units). The two remaining items also use a Likert scale but with 3 categories ordered from 0 to 2. The responses from participants allows them to be classified into three levels: without risk of dependence, at risk consumption and probable alcohol dependence syndrome (ADS) (see cut-off points in data analysis section). The recent validation of the test on Spanish university population by García, Novalbos, Martínez and O’Ferrall [43] found the internal consistency of the AUDIT test to be α = 0.75. In our sample the internal consistency of the test was α = 0.74.

Anxiety: to measure levels of anxiety we used the State-Trait Anxiety Inventory (STAI) adapted by Buela-Casal, Guillén-Riquelme and Seisdedos-Cubero [44]. This questionnaire is composed of 40 items which evaluate two different concepts: anxiety as a state (a transitory emotional response) and anxiety as a trait (a constant anxious condition). The 20 items of each subsection use a Likert scale of 4 categories from 0 (almost never) to 3 (very often/almost always). The analysis of psychometric properties of the instrument for university population showed a high internal consistency (α = 0.93) [45]. In our sample, the internal consistency was α = 0.58.

Emotional Eater: to evaluate this variable the Emotional Eating Questionnaire was used (EEQ) [46], consisting of 10 items on a 4-category Likert scale from 0 (never) to 3 (always). The test consists of 3 subscales which measure the lack of control (disinhibition) in eating, food preferences (with high calory content) and feelings of guilt. Additionally, the test provides a global score to distinguish between the non-emotional eater, the slightly emotional eater, the emotional eater and the highly emotional eater. In this last category there is a risk of developing an eating disorder. The temporal stability shows medium–high correlation in the test-retest average (*r* = 0.702; *p* < 0.001) and the internal consistency of the subscales range from α =0.61 to α = 0.77. This indicates the appropriate reliability of the instrument [46]. In our sample, the internal consistency was α = 0.79 and the subscales varied between α = 0.58 and α = 0.73.

### 2.3. Procedure

The participants were contacted during the 2019–2020 academic year by a professor of their degree who, after explaining the purpose of the study and the confidentiality of the data, provided an informed consent form to those interested in participating. In no case did the participants receive any compensation for taking part in the test and their participation was entirely voluntary and disinterested. The questionnaires fulfillment lasted about 30 min. Data collection was carried out on paper and in person between September 2019 and January 2020. The study fully complied with the Helsinki Declaration and was approved by the Research Ethics Committee of the University (17/2019). The evaluation protocols were provided physically, and the data was uploaded onto a database for analysis using SPSS software (SPSS Inc., Chicago, IL, USA).

### 2.4. Data Analysis

We used IBM SPSS version 22 software for data analysis. Given that the first objective of the study was descriptive, a descriptive analysis was made of all the variables. For adherence to Mediterranean diet we used the limits marked by Serra [13] considering that 3 or less points is considered as poor Mediterranean diet, between 4 and 7 points is considered to be average but needs to improve Mediterranean diet, and 8 or more as optimum Mediterranean diet.

For alcohol consumption we used the limits established by García Carretero [43] for the interpretation of the AUDIT score among Spanish university students (different cut-offs for men and women: 8 for men and 6 for women for high risk consumption and a score of 13 for both men and women to consider probable ADS). Regarding state-anxiety, the first and third quartile were used as points of reference to differentiate between anxiety levels.

Regarding the emotional eater, we used the cut-offs established by Garaulet [46], considering 5 or less points as non-emotional eater, between 6 and 10 points as slightly emotional eater, between 11 and 20 points as emotional eater, and between 21 and 30 as highly emotional eater.

The differences in these variables based on sex were analyzed using Student’s *t*-test and the university degree using a one-way ANOVA test. For the second objective, correlational and predictive, the matrix of correlations between variables was determined using the Pearson correlation coefficient. Subsequently, multiple linear regressions were made using as criteria the degree of adhesion to the Mediterranean diet, alcohol consumption, and level of emotional eating and included the predictive variables which showed significant correlation to the criteria.

## 3. Results

### 3.1. Descriptive Analysis of the Degree of Adherence to the Mediterranean Diet, Alcohol Consumption, Anxiety, and Emotional Eating

The results of the questionnaire regarding AMD show that 20.7% of participants have a poor diet (*N* = 52), while 63.7% show dietary habits that need improvement (*N* = 161) and only 15.5% have an optimum Mediterranean diet (*N* = 39). In analyzing the differences according to sex and degree (Table 2 and Table 3), differences were only found in relation to the degree, although the effect size is low. Specifically, students of physiotherapy showed higher levels of AMD than students of nursing and psychology (*p* = 0.005; *p* = 0.001, respectively).

Secondly, regarding alcohol consumption it was found that 80% of the sample may be considered as not at risk of developing alcohol dependency (*N* = 201), 17.6% are at risk of developing alcohol dependency (*N* = 45) and 2.4% can be considered as having alcohol dependency (*N* = 6). As shown in Table 2 and Table 3, no differences were found in alcohol consumption according to sex or university degree.

Regarding anxiety, the results show that 25.2% of the sample experience low levels of state-anxiety (*N* = 63) compared to 51.2% who report moderate (*N* = 129) and 23.6% who report high levels (*N* = 60). No differences were found in levels of state-anxiety according to sex but there were differences according to university degree with a low effect size (see Table 2 and Table 3, respectively). Pairwise comparisons show that students of psychology experience higher levels of state-anxiety than nursing or physiotherapy students (*p* < 0.001).

With regards to trait-anxiety, the percentage of participants with low, medium, and high levels are similar to figures for state-anxiety, 24.8%, 57.5%, and 17.7%, respectively (*N* = 62, *N* = 145 and *N* = 45, respectively), and no differences were found between men and women (see Table 2). Differences in this variable were only detected according to studies, as shown in Table 3, specifically, among students of psychology and nursing (*p* < 0.001).

Finally, regarding emotional eating, the study found that some 28.2% of the sample can be considered as non-emotional eaters (*N* = 71), some 40.4% are slightly emotional eaters (*N* = 102), 29% are emotional eaters (*N* = 73) and 2.4% are highly emotional eaters (*N* = 6). In analyzing the differences between men and women, statistically significant differences were found in the subscale of guilty and in the global score, where women showed higher figures (see Table 2). The point biserial correlation shows that sex and this variable share 7% variance (*R_XY_* = 0.268; *R_XY_*^2^ = 0.071) for the subscale of guilty and 3% for the global score (*R_XY_* = 0.189; *R_XY_*^2^ = 0.035), that is, a low effect size.

With regard to the differences in emotional eater by degree, as shown in Table 3, statistically significant differences were found only among students of different degrees in the subscale of food preferences, with a low effect size. Pairwise comparisons reveal differences between students of psychology and nursing in favor of the former (*p* = 0.015).

### 3.2. Correlational Analysis and Predictive Model of Adherence to the Mediterranean Diet (AMD), Alcohol Consumption, and Emotional Eating

As for the second objective of the research project, Table 4 indicates the bivariate correlations using the Pearson correlation coefficient for the variables of the study. As can been observed, levels of state-anxiety show a direct linear relation with all subscales of the Emotional Eater, while for trait-anxiety this is only the case in the subscales of food preference and disinhibition. Additionally, there are also significant direct linear correlations, *p <* 0.05, between the subscale guilty and alcohol consumption and trait-anxiety and AMD.

To obtain predictive models for the different subscales of Emotional Eater, AMD and alcohol consumption, anxiety (state and trait), sex, and age were included as independent variables. For all of these, multiple linear regressions were made using forward selection.

For the subscale of food preference, state-anxiety proved to be the only predictive variable (β = 0.037; *p* < 0.001), which explains 13% of the variance in the response variable (*F* = 40.099; *p* < 0.001). State-anxiety was also the only predictive variable in the subscale disinhibition (β = 0.092; *p* < 0.001), explaining 12% of the variance (*F* = 36.931; *p* < 0.001). On the contrary, for the subscale guilt, in addition to state-anxiety (β = 0.029; *p* = 0.001), the variables of sex (β = 0.1.004; *p* < 0.001) which resulted the most predictive variable. The resulting model explains 11% of the variance in the variable (*F* = 16.382; *p* < 0.001). Sex and state-anxiety results were also predictive of a global score of the EEQ, but in this case, the anxiety was a greater predictive variable (β = 0.156; *p* < 0.001) instead of sex (β = 1.858; *p* = 0.004). The model was statistically significant (*F* = 30.223; *p* < 0.001) and explains 19% of the variance in the variable.

As for the predictive model for AMD, only trait-anxiety was a predictive variable (β = 0.089; *p* = 0.017), explaining 1.9% of the variance in the variable (*F* = 5.749; *p* = 0.017). By contrast, for alcohol consumption, no entered variable was a significant predictor.

In all the multiple regressions the Durbin–Watson statistic showed optimum values (1.81–2.11), as was also the case for tolerance levels (0.92–1.00) and the variance inflation factor (1.00–1.07), which ensures compliance with the supposed independence of residuals and collinearity.

## 4. Discussion

For the first objective of the study, an evaluation was made of the eating habits, alcohol consumption, degrees of emotional eating, and anxiety (state/trait) among the students. With regards to students’ eating habits, the KIDMED index measuring AMD shows that a large proportion of participants have a poor diet (20.7%) or have eating habits that need improvement (63.7%). These values are similar to those found in previous studies carried out among university Spanish students in Madrid, Navarra, and Granada [5,11,14,47,48]. Some authors have pointed to a clear trend among Spanish young people in abandoning the Mediterranean diet and lifestyle [49]. Physiotherapy students showed higher levels of AMD than nursing or psychology students. This may be related to the higher levels of physical activity of physiotherapy students: studies of Spanish university students indicate that AMD is higher among those who are more physically active and less sedentary [11,14,50]. No significant differences were found according to sex, in line with the AMD results of KIDMED testing among large samples of Spanish young people [13,42].

The physical and cognitive benefits of the Mediterranean diet have been demonstrated among countries of the Mediterranean basin including Spain [10,11,15,47,51]. It is, therefore, proper to promote AMD among the general public and particularly among future health professionals, such as students of nursing, physiotherapy, and psychology, through information and education programs, some of which have demonstrated their effectiveness [47].

Regarding alcohol consumption, some 80% of participants can be considered as not at risk of problematic alcohol consumption, although 17.6% show results that indicate a risk of dependency and 2.4% can be considered as having alcohol dependency syndrome (ADS). These results, while worrying, are not as alarming as those found in previous research which show higher rates of alcohol consumption among university students and a higher percentage of participants with ADS [8,52,53], which is in line with the findings of the Encuesta Nacional sobre Drogas (National Drug Survey) [54], which points to a fall in alcohol consumption among secondary students in recent years. It has been proposed that alcohol use may be sensitive to contextual factors, suggesting that drinking behavior depends on the time spent in the university: the longer the period a student spends in the university, the higher his/her risk of drinking [19]. It is possible that, nowadays, college students generally spend a shorter period at university, due to the use of online platforms for many activities and virtual class attendance on the internet.

No differences were detected in the consumption of alcohol between men and women, which confirms that alcohol consumption can no longer be largely associated with men, as previous studies have shown, and is now equally prevalent in both sexes [18,55,56,57]. Neither were any differences found in the consumption of alcohol according to area of study, which seems to indicate that, among students of degrees within the field of health sciences, the choice of future profession has no bearing on alcohol consumption. In fact, no research has found differences in alcohol consumption among students of different degrees, nor differences between degrees in the field of health sciences, which points to possible avenues for future research. Perhaps this is another explanation for the lower rates of alcohol consumption found in this study in comparison with previous research about alcohol consumption among university students: all participants were enrolled in health science courses, and are possibly more aware of health issues.

The findings related to anxiety show that some 25.2% of participants experience low levels of state-anxiety, as opposed to 51.2% who reported moderate or 23.6% who reported high levels of state-anxiety; with regards to trait-anxiety, the percentage of participants showing low, moderate, and high levels are similar to the figures for state-anxiety (24.8%, 57.5%, and 17.7%, respectively). The fact that over half the sample show moderate levels of anxiety, both state and trait, is in line with other studies which have found higher levels of anxiety among university students compared to the general population [30]. No significant differences were found between men and women for these two measures of anxiety, but differences were found in terms of university degree: students of psychology experience higher levels of anxiety (both state and trait). Previous studies have detected generally higher levels of anxiety among students of health sciences, such as medicine [58] or nursing [59], but no studies have been found which compare the different degrees in relation to anxiety. Identifying degrees (as in this case, psychology) where students show higher levels of anxiety can help develop and implement effective coping strategies for stress, often associated with academic demands [31,32]. High anxiety can lead to physical, emotional, and mental health problems, and has also been associated with lower academic performance [60].

In relation to emotional eating, the study shows that 28.2% of the sample can be classified as non-emotional eaters, some 40.4% as slightly emotional eaters, 29% as emotional eaters and 2.4% as highly emotional eaters. These results show that almost a third of participants exhibit this pattern of eating behavior with a certain frequency. Some authors have related this type of eating behavior with feelings of anxiety [26], while others argue that boredom is a more significant factor than anxiety or depression [25]. An analysis of the differences between men and women in the various subscales shows differences only in the subscale of guilt, where women score higher. Studies analyzing sex differences with regard to emotional eating have found certain differences: among men, emotional eating is associated principally with symptoms of depression, while in women it is principally linked to feelings of anxiety [27]. It would appear that the mechanisms of emotional eating among men and women are different and further study is needed into the factors triggering emotional eating among men and women. Regarding the differences found in terms of area of study, findings show that psychology students experience higher levels of anxiety compared to other degrees, and also score above average in the subscale of food preferences.

Regarding the second objective, relating levels of anxiety to AMD, emotional eating, and alcohol consumption, some interesting connections have been found. Levels of state-anxiety reveal a direct linear relation with all subscales of the emotional eating questionnaire (disinhibition, food preferences and guilt), while trait-anxiety is related only to the subscales of disinhibition and food preferences. This is coherent with findings of previous studies which have related the increase in emotional eating with anxiety and stress [61]. Additionally, the relationship between anxiety and increased intake of high-calory foods as a coping strategy [22] can alter the cerebral reward system, particularly at a stage of life coinciding with the end of the cerebral maturation period [62], with significant long-term repercussions as the intake of unhealthy food can become an important reinforcer [23]. Regarding the difference between state anxiety and trait anxiety, a recent study of adult women showed a linear relationship between emotional eating and anxiety, particularly for trait anxiety [63]. More study is needed into the relationship between anxiety and eating behavior. In this study both state-anxiety and trait-anxiety are related to disinhibition and food preferences, but only state-anxiety is related to guilt. State-anxiety is more associated with specific external situations and is more susceptible to change than trait anxiety, by changing stressful contextual factors or implementing appropriate coping strategies. It is important to take these differences into account when designing prevention/intervention strategies and profiles of unhealthy eating habits.

Although poor eating habits have long been associated with alcohol consumption among university students [64,65], the present study found no relation between the level of alcohol consumption and the degree of AMD. However, these results are coherent with the notion that both poor eating habits are coping strategies, normally exclusive, adopted by university students in situation of stress [37]. This points to the need for university students to improve their coping strategies in favor of more active and effective strategies. With regard to emotional eating, a relationship was found only between the subscales of guilty and alcohol consumption. The propensity for feelings of guilty (in general and not only in relation to specific eating habits) has been associated with substance addition [66], including alcohol abuse [67].

Finally, the study found a direct relationship between the level of trait anxiety and AMD. An obsession with healthy eating has been associated with personality disorders, low self-esteem, and anxiety problems [68,69], which may partly explain these results. It has been found that a direct relationship exists between obsessive preoccupation with healthy food and perfectionism [70], and anxiety seems to be a mediator of the relationship between perfectionism and eating disorders [71].

Regarding the third objective of the study, to determine which factors (sex, age, and anxiety levels) are predictive of alcohol consumption, emotional eating and AMD, state-anxiety proved to be the most predictive for the subscale of food preferences and disinhibition on the questionnaire of emotional eating. As mentioned above, increased intake of high-calory food has been identified as a coping strategy for anxiety [22] and the relationship between anxiety and disinhibition has been established in many studies dealing with problematic eating habits, although from a clinical perspective [72,73]. It would be interesting to determine if difficulties in impulse control associated with emotional eating and immediate gratification, which ignore long-term negative consequences, only occur in the context of diet, and not in other scenarios such internet use (gaming or social media), or in everyday contexts. For the subscale of guilt, in addition to state-anxiety, other variables such as sex (women) play a part. In this sense, social pressure suffered by women to achieve a canon of beauty can be an explanation to the sense of guilt after emotional eating. For AMD, only trait-anxiety was a predictor. Further studies are needed in order to deepen how personality traits or stable behavior patterns contribute to adhere to healthier eating habits. Thus, understanding the function of adaptative or maladaptive eating and drinking behaviors may result in a more effective intervention [74,75].

These findings could enable the design and implementation of educational programs to promote healthy habits among university students. It is important that future healthcare professionals’ value and adopt healthy habits and lifestyles given that they will be those primarily responsible for encouraging healthy habits and risk prevention among their patients. In this sense, it seems important to control anxiety levels of the students, particularly state-anxiety levels, for the adoption of unhealthy habits. It is relevant to teach students how to manage with stressful situations and how to implement coping strategies.

This study has several limitations. Firstly, the participants were only from psychology, nursing, and physiotherapy courses, and it would be interesting to count with students from all the degrees of healthcare professions. A more representative sample should be required, as in other studies that partially investigate some of us associations and longitudinal studies are needed.

Secondly, it would be important to compare the results found with those obtained with students from degrees with no relationship with healthcare. Thirdly, it would be interesting to know the grams of alcohol intake on the participants ad correlated with AUDIT, AMD, anxiety, and emotional eating. And finally, the adaptability of the instruments must be improved, and it would have been desirable to include other psychological variables such as depression, which has also shown high prevalence among college population [32]. Further studies in this topic are needed.

Previous work on this area suggests the association between AMD and quality of life and others the relationship between AMD and the wellbeing of Spanish college students [76]. Positive associations between AMD adherence and higher levels of subjective happiness have been found in adolescents [77]. Furthermore, several studies have suggested the effects of a dietary intervention on mental health: a systematic review found improvements in depression scores in eight of the seventeen studies reviewed, and in two of the ten studies that measured anxiety, when compared with a control group [78]. The importance of diet in mental health, and not just physical health, seems indisputable. However, an important aspect of the current study needs to be highlighted: the first approach to the study of the relationships between AMD, emotional eating, alcohol intake, and anxiety, which can contribute to the start of a line of research that can be fruitful in health care.

## 5. Conclusions

The results of this study indicate about the need to improve nutritional habits between college students, without significant differences according to sex. Regarding alcohol intake, most of participants can be considered as not at risk of problematic alcohol consumption, although the fifth part of the sample showed a problematic alcohol intake, suggesting the need of reduce this consumption among students. No differences were detected between men and women. Although poor eating habits have long been associated with alcohol consumption among university students, the present study found no relation between the level of alcohol consumption and the degree of AMD.

Half of the student showed moderate levels of anxiety, both state and trait, without differences men and women, although differences were found in terms of university degree: students of psychology experience higher levels of anxiety (state and trait).

About emotional eating, this study shows that almost a third of participants exhibit this pattern of eating behavior with a certain frequency. No differences were found between men and women either, except in the subscale of guilt, where women scored higher. Some interesting connection has been found between anxiety and emotional eating: state-anxiety reveal a direct linear relation with all subscales of the emotional eating questionnaire (disinhibition, food preferences and guilt), while trait-anxiety is related only to the subscales of disinhibition and food preferences. Moreover, it has been found a direct relationship between the level of trait anxiety and AMD.

Finally, state-anxiety proved to be the most predictive for the subscale of food preferences and disinhibition on the questionnaire of emotional eating. For the subscale of guilt, in addition to state-anxiety, other variables such as sex (women) played a part. For AMD, only trait-anxiety was a predictor.

## Figures and Tables

**Table 1 nutrients-12-02224-t001:** Descriptive analysis of age by sex and area of study.

**Degree**	**Sex**	***N* (%)**	**M**	**SD**
Nursing	Men	20 (16.4)	20.55	3.01
	Women	102 (83.6)	21.66	6.28
	Total	122 (48.41)	21.46	5.83
Physiotherapy	Men	25 (42.4)	21.64	2.73
	Women	34 (57.6)	21.00	4.18
	Total	59 (23.41)	21.22	3.61
Psychology	Men	16 (22.5)	21.44	3.68
	Women	55 (77.5)	21.53	3.14
	Total	71 (28.17)	21.51	3.24
Total	Men	61 (24.2)	21.23	3.08
	Women	191 (75.8)	21.51	5.18
	Total	252	21.42	4.73

M = average; SD = standard deviation.

**Table 2 nutrients-12-02224-t002:** Descriptive statistics and group contrasts by variables according to sex.

Measures/Sex	M	SD	T Student	*p*
Mediterranean diet			−0.326	0.744
Men	5.36	2.83		
Women	5.72	8.49		
Total	5.62	7.49		
AUDIT			1.579	0.118
Men	4.90	4.36		
Women	3.94	3.28		
Total	4.14	3.58		
State Anxiety			−0.772	0.441
Men	21.5	11.23		
Women	22.91	12.74		
Total	22.51	12.34		
Trait Anxiety			−1.329	0.185
Men	19.81	10.63		
Women	22.31	13.39		
Total	21.63	12.8		
EE: Food preferences			−1.892	0.060
Men	1.75	1.24		
Women	2.09	1.21		
Total	2.01	1.23		
EE: Guilty			−5.358	<0.001 *
Men	1.11	1.15		
Women	2.15	1.72		
Total	1.89	1.65		
EE: Disinhibition			−1.578	0.116
Men	4.19	3.17		
Women	4.92	3.13		
Total	4.75	3.15		
EE: Global score				
Men	7.06	4.62	−3.048	0.003 *
Women	9.17	4.72		
Total	8.67	4.77		

* *p* < 0.05; M = average; SD = standard deviation; EE = Emotional Eater.

**Table 3 nutrients-12-02224-t003:** Descriptive statistics and group contrasts by variables according to degree.

Measures/Degree	M	SD	F-Test	*p*	*η* ^2^ *p*
Mediterranean Diet			0.703	0.001 *	0.053
Nursing	5.00	2.55			
Physiotherapy	8.66	14.58			
Psychology	4.14	1.47			
AUDIT			0.201	0.818	0.002
Nursing	4.00	3.28			
Physiotherapy	4.25	3.93			
Psychology	4.30	3.81			
State Anxiety			23.448	<0.001 *	0.157
Nursing	18.40	9.77			
Physiotherapy	22.16	12.17			
Psychology	29.98	13.16			
Trait Anxiety			9.99	<0.001 *	0.073
Nursing	18.51	10.69			
Physiotherapy	22.05	14.85			
Psychology	26.63	12.82			
EE: Food preferences			3.928	0.021 *	0.030
Nursing	1.83	1.09			
Physiotherapy	2.03	1.27			
Psychology	2.33	1.36			
EE: Guilty			0.582	0.559	0.005
Nursing	1.91	1.62			
Physiotherapy	1.71	1.46			
Psychology	2.02	1.85			
EE: Disinhibition			0.795	0.453	0.006
Nursing	4.50	3.28			
Physiotherapy	4.93	3.08			
Psychology	5.04	2.96			
EE: Global score					
Nursing	8.24	4.80	1.353	0.260	0.011
Physiotherapy	8.68	4.64			
Psychology	9.40	4.79			

* *p* < 0.05; M = average; SD = standard deviation; EE = Emotional Eater.

**Table 4 nutrients-12-02224-t004:** Bivariate correlations of the variables of the study.

Measures	1	2	3	4	5	6	7	8
1. Mediterranean Diet	-	0.009	0.012	0.154 *	−0.002	0.025	0.062	0.049
2. AUDIT		-	0.052	0.019	0.026	0.141 *	0.113	0.130
3. State Anxiety			-	0.609 **	0.361 **	0.226 *	0.349 **	0.402 **
4. Trait Anxiety				-	0.203 **	0.089	0.176 **	0.200 **
5. EE: Food preference					-	0.212 **	0.465 **	0.639 **
6. EE: Guilt						-	0.394 **	0.662 **
7. EE: Disinhibition							-	0.917 **
8. EE: Global score								-

* *p* < 0.05; ** *p* < 0.01; EE: Emotional Eater.

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
