# Peer review of "Are Adherence to the Mediterranean Diet, Emotional Eating, Alcohol Intake, and Anxiety Related in University Students in Spain?"

_nutrients, 2020, doi:10.3390/nu12082224_

Round 1
Reviewer 1 Report
General Comments
This study reports the prevalence of Mediterranean diet adherence, emotional eating, alcohol intake and anxiety in university students in Spain. It examines associations between each variable, and whether age, sex, anxiety and area of study are significant predictors.
The manuscript is well-written and provides some insight into psychological constructs and health behaviours in university students. However, the rationale for the study design and analyses is somewhat unclear. Further detail is needed to justify why this study is needed, and why each comparisons is necessary.
Abstract
Page 1, Line 16: Please define abbreviations and what each test measures.
Page 1, Line 21-22: What was the nature of these associations? Please provide coefficients and p values for each.
Page 1, Line 22-23: The sentence regarding predictive models is unclear. Which predictors and outcomes do these values relate to?
Introduction
Page 1, Line 31-33: This statement requires a reference.
Page 1, Line 33: Please provide a definition of the Mediterranean diet.
Page 1, Line 33-34: The statement that “unhealthy behaviour is associated with a departure from the Mediterranean dietary model” is specific to Mediterranean populations, where the diet is adhered to at a higher rate than non-Mediterranean countries. Please consider rephrasing.
Page 2, Line 19: Phrase should be "feelings of guilt".
Page 2, Line 23-26. Please consider revising the statement “Stress, anxiety and depression appear to be the principal psychological problems experienced during university studies all over the world and studies show university students have poorer mental health than the general population”. The cited study, limited to an Italian sample, did not measure stress, anxiety or depression, and did not compare the mental health of university students to the general population.
Page 2, Line 43-47. The study objectives are somewhat unfocused, and further detail is required to justify the need for the study. Why is it necessary to explore all of these associations? It is currently unclear what this investigation offers above previous studies, and how the results can be used to develop prevention and intervention strategies.
Page 2, Line 48-52: Objectives should be numbered
Page 2, Line 50: Justification is needed for exploring differences between students of physiotherapy, psychology and nursing backgrounds? Is there an expectation that there will be differences between these cohorts? If so, why? Likewise, it is unclear how age will be related to each outcome.
The authors have used the term ‘gender’ (which is a psychosocial construct) to define men and women, whereas sex (which is an anatomical, physiological, or biological phenomenon) may be more appropriate. If so, please change your terminology. On the other hand, if you truly assessed gender please define how this was done and what elements you used to describe this complex variable.
Methods
Table 1: Column labels are unclear. Do Mean and SD represent age?
Page 3, Line 14-19: Please elaborate on the dietary information captured by the KIDMED. What elements of the Mediterranean diet are assessed?
Page 3, Line 20: Please define the acronym ‘AUDIT’.
Page 4, Line 6: Phrase should be "feelings of guilt".
Procedure: Please outline the procedure of how and when data was collected.
Data Analysis: As there were no specific hypotheses regarding associations with age and sex, consider including these variables as covariates, rather than predictors in regression analyses.
Results
Cut-offs for the interpretation of each test should be reported in Data Analysis rather than Results.
Page 5, Line 17-20: Consider revising this section. Significant differences were also found for AMD, State Anxiety, and Trait Anxiety. While these are reported above, it is misleading to state that “statistically significant differences were found only among students of different degrees in the subscale of food preferences”.
Analyses between groups are only presented for subscales of Emotional Eating, but not for the total score. What were the associations with the Global Emotional Eating score?
Please report the results of all multiple regressions mentioned in your objectives
Page 7, Line 14: It is unclear why the subscale ‘guilty’ was examined as a predictor of alcohol consumption, as this is not consistent with the study objective “To determine if age, gender, anxiety and area of study (physiotherapy, psychology and nursing) are predictors of alcohol consumption, emotional eating and adhesion to the Mediterranean diet.”
Discussion
It is noted that rates of alcohol consumption are lower than previous studies. What are some potential explanations for the lower rates of alcohol consumption in the current study?
Page 7, Line 48: Please consider revising the statement “the choice of future profession has no bearing on alcohol consumption”. As correctly identified by the following sentence, all students were enrolled in health science courses. In comparison to previous studies reporting higher consumption, it is possible that future profession is associated with alcohol consumption.
Please consider addressing the impact of high levels of anxiety in university students.
Page 8, Line 29: It is unclear why psychology is highlighted as being ‘majority women’ when the analyses show no difference between men and women for State and Trait Anxiety.
Page 8, Line 48-49: Please consider revising the statement “However, these results are coherent with the notion that both poor eating habits and excessive alcohol consumption are coping strategies” as alcohol consumption was not associated with State or Trait Anxiety.
Further discussion regarding the difference between State and Trait anxiety, and how these differences relate to results would be beneficial.
Page 9, Line 3-5: Are there any other potential explanations for the positive association between Trait Anxiety and AMD?
Page 9, Line 9: Spelling of “calorie”.
Page 9, Line 12-14: Please clarify which other relevant contexts that should be explored.
Page 9, Line 15-17: Please elaborate on these results and how they relate to the literature.
It would be of benefit to discuss how these findings could be used to inform the design and implementation of educational programs to promote healthy habits among university students.
The study’s limitations have not been addressed in sufficient detail. Please acknowledge potential limitations of the study design, including the use of a cross-sectional analysis limited to health science students.
Reviewer 2 Report
The dates are not well set, they are all in the same square bracket, for example, the second quote would be: [2-5]
METHODS
Participants: It is not a heterogeneous sample in terms of gender, since it was for convenience, this could have been taken into account to avoid possible biases in results,as well as increasing the sample size.
- Add inclusion criteria for sample selection.
- In table 1 add the percentages to the total.
Variables and instruments:
- Of the three instruments used (KIDMED, AUDIT and STAI), the scoring range must be added for the results, for example in KIDMED in which range is a poor diet, in which the need to improve eating habits and in which one optimal Mediterranean diet.
Process:
- Indicate how long it took to fill in the questionnaires.
Statistic analysis:
- Detail and develop this section better: specify the version of the SPSS (company, city, state abbry, if USA, country), the use of factor loads, the explained variance, everything related to the regression.
RESULTS
I suggest add another table for the second objective for the significant predictive correlation, adding the variances in the table, it would be better understood.
DISCUSSION
- Add the limitations of the study.
- I suggest reviewing these articles that may be valuable for the discussion section:
López-Olivares, M.; Mohatar-Barba, M.; Fernández-Gómez, E.; Enrique-Mirón, C. Mediterranean Diet and the Emotional Well-Being of Students of the Campus of Melilla (University of Granada). Nutrients 2020, 12, 1826.
Ferrer-Cascales, R.; Albaladejo-Blázquez, N.; Ruiz-Robledillo, N.; Clement-Carbonell, V.; Sánchez-SanSegundo, M.; Zaragoza-Martí, A. Higuer Adherence to the Mediterranean Diet is related to more subjective happiness in adolescents: The role of health-related quality of life. Nutrients 2019, 11, 698.
Opie, R.; O’Neil, A.; Itsiopoulos, C.; Jacka, F. The impact of whole-of-diet interventions on depression and anxiety: A systematic review of randomised controlled trials. Public Health Nutr. 2014, 18, 2074–2093.
